# A Study of GUS Expression in *Arabidopsis* as a Tool for the Evaluation of Gene Evolution, Function and the Role of Expression Derived from Gene Duplication

**DOI:** 10.3390/plants12102051

**Published:** 2023-05-22

**Authors:** Leonardo Bruno, Matteo Ronchini, Giorgio Binelli, Antonella Muto, Adriana Chiappetta, Maria Beatrice Bitonti, Paolo Gerola

**Affiliations:** 1Dipartimento di Biologia, Ecologia e Scienze della Terra, Università della Calabria, Arcavacata di Rende, 87036 Cosenza, Italy; antonella.muto@unical.it (A.M.); adriana.chiappetta@unical.it (A.C.); b.bitonti@unical.it (M.B.B.); 2Dipartimento di Scienze Teoriche e Applicate, Università degli Studi dell’Insubria, 21100 Varese, Italy; matteoronc@gmail.com (M.R.); paolo.gerola@gmail.com (P.G.); 3Dipartimento di Biotecnologie e Scienze della Vita, Università degli Studi dell’Insubria, 21100 Varese, Italy; giorgio.binelli@uninsubria.it

**Keywords:** gene duplication, GUS genes, evolutionary tree, expression pattern, function change

## Abstract

Gene duplication played a fundamental role in eukaryote evolution and different copies of a given gene can be present in extant species, often with expressions and functions differentiated during evolution. We assume that, when such differentiation occurs in a gene copy, this may be indicated by its maintenance in all the derived species. To verify this hypothesis, we compared the histological expression domains of the three β-glucuronidase genes (*AtGUS*) present in *Arabidopsis thaliana* with the GUS evolutionary tree in angiosperms. We found that *AtGUS* gene expression overlaps in the shoot apex, the floral bud and the root hairs. In the root apex, *AtGUS3* expression differs completely from *AtGUS1* and *AtGUS2*, whose transcripts are present in the root cap meristem and columella, in the staminal cell niche, in the epidermis and in the proximal cortex. Conversely, *AtGUS3* transcripts are limited to the old border-like cells of calyptra and those found along the protodermal cell line. The *GUS* evolutionary tree reveals that the two main clusters (named *GUS1* and *GUS3*) originate from a duplication event predating angiosperm radiation. AtGUS3 belongs to the GUS3 cluster, while *AtGUS1* and *AtGUS2*, which originate from a duplication event that occurred in an ancestor of the Brassicaceae family, are found together in the *GUS1* cluster. There is another, previously undescribed cluster, called GUS4, originating from a very ancient duplication event. While the copy of GUS4 has been lost in many species, copies of GUS3 and GUS1 have been conserved in all species examined.

## 1. Introduction

Whole-genome duplication events (WGD) constitute fundamental processes in gene family formation and evolution in eukaryotes [1,2] and their relevance in shaping genome architecture and new functionalization has been underlined, particularly in plants [3,4].

The duplication of genome content can occur due to mechanisms which are different, but not mutually exclusive, leading to the duplication of single genes, chromosome segments (segmental duplication) or WGD [2,5]. Anomalies of crossing over are often involved in cluster tandem duplications, while chromosomal anomalies lead to segmental duplications [2]. Transposons are also involved in the capture and transportation of gene copies (transduplication) either (i) by intrachromosomal recombination or (ii) by genomic reintegration of reverse-transcribed 49 cDNAs (retropositioning).

However, polyploidy associated with WGD events has been proposed as the most important contributor to gene duplication and is considered to be the major source of the huge genomic variability and plasticity that drive plant evolution [6].

Several WGD have been proposed as occurring during seed plant evolution. Aside from a WGD event that is difficult to date but which is believed to have taken place ca 350 Mya and to have been shared by all existent seed plants, the most ancient one occurred ca 200 Mya, after gymnosperm diversion, before angiosperm radiation [6]. WGDs subsequently occurred during angiosperm evolution, ranging from the ancient hexaploidization event in the eucotyledon ancestor (70–133 Mya) to those that occurred in the ancestors of families such as Poaceae, Solanaceae, Brassicaceae, Fabaceae, and Asteraceae, (24–70 Mya), up to the most recent ones that were specific to minor phyla, like that which occurred in *Gossypium* after *Theobroma* divergence (3–20 Mya) [4,6,7,8,9,10,11,12,13,14,15,16,17,18,19].

Different fates can follow a polyploidization event. The success of a newly formed polyploid species can be impaired by genomic instability due to both genetic and epigenetic factors. Later on, the copies of many genes can be lost, thus leading to a reduction in genomic redundancy and to “diploidization” (fragmentation process) [20]. However, although the large majority of duplicated genes are lost in a process called pseudogenization [18,21], genes can be retained and often acquire novel and/or expanded functionalities.

Gene conservation, particularly in the case of developmental and regulatory genes, can be influenced by dosage effects, i.e., increased production of beneficial gene products and dosage balance constraints) [22,23,24]. The following classes of genes have been reported to be preferentially retained following WGD: genes encoding members of large multiprotein complexes, transcription factors, and genes associated with large numbers of conserved non-coding regulatory elements [25,26,27,28,29].

“Subfunctionalization” (SF), “neofunctionalization” (NF) and “escape from adaptative conflict” (EAC) are also processes that are not mutually exclusive but which are implicated in gene retention following gene duplication [1,2,14,18,22,23,30,31,32,33].

SF is a mechanism that can take place following the duplication of a gene which has more regulation domains, allowing a differential gene expression in the space (different cells, tissues or organs) or in time (during plant development or plant life and cycle). In fact, unequal loss of the regulation domains in the two gene copies leads to a partition of the ancestral functions, meaning that both copies have to be maintained [2,22,30,31].

NF is the acquisition of a new function by one of the gene copies. This occurs, for example, following the transfer of the gene under different regulation domains. The new function acquired by the gene, when expressed in a different organ or tissue or at a different time, can lead to changes in the coding gene region driven by the positive selection of the gene product according to the new function [2,31].

EAC is a gene conservation mechanism that acts following the duplication of genes coding for proteins with multiple catalytic or structural functions. After duplication, gene copies can each evolve according to the selective pressure of one particular function, escaping from the adaptative conflict that hindered the evolution of the ancestor gene [2].

Regardless of the mechanism involved, the maintenance of gene copies in most or all the 90 taxa descendants from the same ancestor is probably associated with a differentiation in gene expression or function, particularly when following a paleo-polyploidization event [18]. Therefore, differences in gene expression might be hypothesized via gene evolutionary tree analysis. In the present work, to verify such possibility, we planned to compare the evolutionary tree of β-glucuronidases genes (GUS) and their expression pattern in Arabidopsis thaliana [34].

*GUS*s encode glycosyl-hydrolases (GHs), enzymes that catalyse the hydrolysis of the glycosidic bond between glucuronic acid and other carbohydrates or molecules different from sugars termed aglycones [35]. A very high number (85) of GHs families have been described in *Arabidopsis* [36] demonstrating the importance of the enzymes involved in the synthesis, modification, and breakdown of carbohydrates, also termed CAZymes (carbohydrate-active enzymes). GUSs have been identified in all living organisms and classified into three families: GH1, GH2 and GH79 based on their amino acid sequence and on their stereochemistry [37,38,39]. GUSs, belonging to the GH79 family, are widely distributed in plants [40], while GH2 is absent [35]. A GUS sequence was determined for the first time in *Scutellaria baicalensis* [41] and three different *GUS* genes have been identified in *Arabidopsis thaliana*: *AtGUS1*, *AtGUS2* and *AtGUS3* [42]. Further studies have confirmed the wide diffusion of GUSs in plants [35,43,44,45,46,47] and the presence of enzymatic activity in all the different organs [40,48,49,50,51,52,53,54,55,56,57,58,59].

Concerning the GUS role in plants, their involvement in the changes of polysaccharide moieties [40,60] or in the release of signal molecules [50,57,61,62,63] has been proposed. In particular, it has been demonstrated that GUSs are involved in regulating the glycosylation degree of arabinogalactan proteins [60] and in the changes in cell wall composition associated with cell elongation [40,60]. In modulating signal molecules, GUS works in opposition/coordination to β-glucuronosyltransferase (UGTs) enzymes; these add glucuronic acid to different molecules, causing their physiological inactivation [42,62]. This was demonstrated in *S. baicalensis*, where baicalein presence is under the coordinate control of UGT and GUS activities [57,63]. An involvement of UGT-GUS antagonistic function has been also envisaged in relation to cell cycle regulation in the root cap meristem. In particular, it has been proposed that, in *Pisum sativum*, UGT (PsUGT1) plays a key role in promoting cell division through glycosylation and the consequent inactivation of a mitosis inhibitor, probably a flavonoid, which is instead released by GUS with consequent cell cycle inhibition [61,62]. Consistently with this assumption, inhibition of GUS activity through saccharolactone causes an enhanced production of border-like cells in the root apex of *A. thaliana*, indicating an increase in the mitotic activity of root cap meristem [61]. Moreover, in several species (pea, alfalfa and *A. thaliana*) uidA expression under the PsUGT1 promoter is lethal [61].

Despite the fact that the time- and space-dependent modulation of gene expression is essential in the control of developmental processes, most of the data on the cyto-histological domains of endogenous GUS expression in plants are inadequate, being mainly based on semiquantitative PCR analysis or GUS activity detection [48]. More recently, through multiprobe in situ hybridization (MISH), we obtained a precise in situ localization of *AtGUS1* and *AtGUS2* expression in *Arabidopsis* apical root region [34]. Co-expression of the two genes was observed in the cap central zone (columella), in the root cap meristem, in the stamina cell niche and in the cortical layers of the proximal meristem. Both genes were not expressed in the cup peripheral layers and in the stelar portion of proximal meristem. These results strongly supported an identical tissue-specific expression for both genes.

In this context, the present work was planned to elucidate the potential new functional role of the *AtGUS3* gene product as observed through the evolutionary tree and provide relative evidence of this new function by MISH. The in situ experiment revealed exclusive expression patterns in the root cap for *GUS3* that we hypothesize may lead to the acquisition of new functions in root development.

## 2. Results

### 2.1. Tissue Specific Expression of GUS Genes in Arabidopsis thaliana

To verify the tissue-specific expression of AtGUS genes, we applied a whole mounting multiprobe in situ hybridization (MISH) technique [34]. This allows researchers to compare closely juxtaposed domains of gene expression inside the different organs and tissues.

AtGUS1, AtGUS2 and AtGUS3 expression was analysed in the shoot vegetative apex, in the flower bud, in the root hairs and in the root apex. The obtained results show that in the shoot apex the expression pattern of three GUS genes is quite similar, hybridization signal being detected in the apical dome (arrowhead), leaf primordia and emerging leaflets (arrow), albeit at a low level (Figure 1).

A weak signal was also observed for GUS3 in the provascular strand, while it was stronger for GUS1 and GUS2. In the developing inflorescence (Figure 2), the presence of GUS1, GUS2 and GUS3 transcripts was higher than in the vegetative shoot apex according to in silico and PCR analysis (data not shown). GUS1 signal was particularly evident in floral organ blastozone (arrow), an area where cell proliferation was active.

Overlapping of GUS1, GUS2 and GUS3 expression was also observable in the root hairs (Figure 3), while a different picture was observable in the root apex, where the expression domains of GUS3 strongly differed from those of GUS1 and GUS2 (Figure 4).

In particular, *GUS1* and *GUS2* transcripts were detected in the staminal cell niche, in the epidermis, in the cortical cell layers of proximal meristem and in the root cap meristem. No expression was detectable in the proximal stele meristem (Figure 4a,b).

In the calyptra, they were expressed in the central zone and only faintly in the cap outermost layers. By contrast, GUS3 transcripts were exclusively detected in the border-like cells in the older calyptra region and along the protodermal cell line (Figure 4c,d). 

Globally, the obtained results showed a similar cytoistological expression domain in both SAM and root hair for all GUSs; conversely, for GUS3 in the root apex we only found the exclusive expression in the border-like cells in the older calyptra region and along the protodermal cell line.

### 2.2. GUS Duplications and Its Evolutionary Tree

The GUS evolutionary tree in angiosperms is shown in Appendix A. *GUS* sequences from Briophyta, (*Physcomitrella patens*), and Lycophyta (*Selaginella moellendorfii*) were used as an outgroup to root the tree. The most evident feature of the tree was the clustering of *GUS* genes into three main groups, each one including the four main phyla of angiosperms, i.e., ancestral angiosperms, monocots, basal eudicots, and eudicots. This subdivision of *GUS* genes in the different clusters and phyla is clearly evident in Figure 5. As shown in Appendix A, two main clusters were present: cluster 3 on one side and another one, further subdivided in the clusters 1 and 4, on the other. The name of the clusters was related to the distribution of *AtGUSs*: *AtGUS3* in cluster 3, *AtGUS1* and *AtGUS2* in cluster 1. No *AtGUS* gene was present in cluster 4 (Appendix A).

WGD events have been suggested as possible causes of some of the gene duplications evidenced by the tree. The presence of Amborellales (ancestral angiosperms) in all the clusters (Figure 5) suggests the occurrence of two GUS duplication events or a hexaploidization one preceding angiosperm radiation; these are probably associated with the WGD events that occur in the seed plants and in the angiosperm ancestors [18].

The presence of two eudicot phyla (named 3.1 and 3.2) in cluster 3 (Figure 5), diverging after basal eudicots, suggests a gene duplication probably associated with the hexaploidization event that occurred in the eudicots ancestor after basal eudicot divergence. Traces of such a WGD event are not deductible in eudicots 1 and 4 clusters, indicating that the loss of the *GUS1* and *GUS4* copies originated from such a WGD occurrence in the eudicot ancestor.

GUS duplications, observable in both eudicot and eonocot phyla, can be associated with WGD events that occurred in family ancestors [18]. This was the case for the duplication events that occurred in Solanaceae (eudicots 1), Brassicaceae (eudicots 1), Fabaceae (eudicots 1 and eudicots 3.2), and Poaceae (monocots 1 and monocots 3): the two GUS copies, originating from the duplication event, evolved as independent genes and were retained in the extant species, giving rise to two identical GUS phyla diverging from the family ancestor. Gene copies that originated from more recent GUS duplications were also evident at both the genus and species level and could be related to known WGD events.

A particular analysis can be performed for the unique GUS sequence (i.e., Baicalanase) available for Scutellaria (Lamiaceae). In all the eudicot phyla, the gene copies of Sesamus (Pedaliaceae), Erythranthe (Phrymacea) and Olea (Oleacea) cluster together, close to Solanales, in agreement with the phylogenetic data; these join the three families in the order of Lamiales, which is evolutionarily related to Solanales. Conversely, the Baicalinase of Scutellaria (Lamiaceae) is located in cluster 3; however, this association diverges very early at the basis of cluster 3.

Besides these considerations from the GUS evolutionary tree, the most interesting observation in relation to this work can be drawn by analysing the distribution of *GUS* genes copies of the different species in the main phyla (Appendix A). While *GUS3.1* and *GUS4* genes are missing in several species or whole families, like Brassicaceae, Cucurbitaceae and Fabaceae (lacking GUS3.1) and Brassicaceae and Asteraceae (lacking GUS4), all the 96 examined species retain at least one *GUS3* and one *GUS1* gene, as occurs if they are associated with specific essential functions.

## 3. Discussion

The aim of this work was to verify tissue-specific expression of *GUS* genes in *Arabidopsis thaliana* and investigate whether the differences in gene expressions were reflected by the angiosperm GUS evolutionary tree. Two copies of the genes were thereafter retained in all the species that diverged after the functional differentiation. Therefore, the presence in the evolutionary tree of two phyla with identical species composition diverging from a common ancestor (“double-gene phylum”) might indicate the occurrence of functional/expressional differentiation in the ancestor before speciation. To simplify, the acronym DFD (duplication plus functional differentiation) can be used to indicate this particular case. The absence of DFD events is likely when the two copies are not both retained in all the evolved species and thus, due to the loss of one of the two copies, the two derived phyla do not include the same species. The more ancient the duplication event and the formation of the “double-gene phylum”, the more likely it is that a DFD event is responsible for the retention of two copies of a gene. However, two other explanations can justify the presence of a “double-gene phylum”. Simply, gene duplication is advantageous despite the fact that the genes maintain the same function/expression, or, especially when the duplication is recent, that there is not enough time to observe the loss of one of the gene copies. Thus, when the duplication event is ancient, the inference of DFD event is more likely although not certain.

*AtGUS1*, *AtGUS2* and *AtGUS3* expression was analysed in plant different organs such as shoot and root apex and flower buds. In order to compare the expression pattern of different *GUS*s, we used the MISH technique [34] that allows the simultaneous detection of different gene transcripts in the same sample. No differences were detected in either shoot or flower buds in the expression patterns of the three *GUS*s. However, a larger signal was detected in the flower bud with respect to the shoot apex.

In particular, higher levels of *GUS* transcripts were observed in the blastozone, perhaps in relation to the very active cell proliferation activity in that region. All three genes also appeared also in the root hairs, supporting the proposed role of *GUS*s in root hair elongation [40].

Unlike what was observed in all the other organs, quite a different expression pattern of *AtGUS3* was observed versus *AtGUS1* and *AtGUS2* in the apical root. *AtGUS1* and *AtGUS2* expression was observed in the root cap meristem and columella, in the staminal cell niche, in the epidermis and in the cortical cell layers of the proximal meristem [34]. This overlapping expression pattern was confirmed in this work: that of *AtGUS3* appeared in a completely different manner to the others, limited to the old border-like cells of calyptra and along the protodermal cell line, where no *AtGUS1* and *AtGUS2* expression was observable. It has to be remarked that *AtGUS3* transcripts were not present in all the border-like cells, but they were limited in the region where they detached from the root. As it has been reported that the presence of glucuronic acid in cell wall polysaccharides is essential for cell–cell adhesion [64,65], we assume that that *GUS*s activity in the old border-like cells is related to the detachment process.

Such relevant differences in the expression and putative function of *AtGUS1* and *AtGUS2* versus *AtGUS3* were somehow reflected in the *GUS* evolutionary tree, which might thus be configured as a suitable tool for inferring events of neo-functionalization (NF), sub-functionalization (SF) or escape from adaptative conflict (EAC).

In favour of the independence of the genes and of the possibility of a recombination event, it can be observed from the genetic map (Appendix A) that the three genes encompass a physical distance of about 10 Mb. Assuming an average genetic-to-physical map ratio across the genome of 2 cM/Mb, the three genes were sufficiently distant from each other to allow recombination to occur, as observed for other species [66].

However, some considerations on the GUS evolutionary tree can be made before addressing its analysis for DFD inference. Phylogenetic information can be already obtained by the analysis of each evolutionary gene tree, particularly when multiple gene copies are present. In our case, the phylogenetic relationships inside the families were mostly in perfect agreement with those reported in the literature and low phylogenetic information was drawn at the higher taxonomic level.

The presence of gene copies must be taken into account when constructing a phylogenetic tree. According to Burleigh (2011) [67], phylogenetic analyses using genome-scale data sets must confront incongruences among gene trees, which in plants is exacerbated by frequent gene duplications and losses. Therefore, we adopted the phylogenetic optimization criterion in which a species tree that minimizes the number of gene duplications induced among a set of gene trees is selected. However, the automatic removal of gene copies can lead to errors, as already reported by Rhufel et al., (2014) [68], where *N. tabacum* appears closely related to *N. sylvestris* and more closely related to *N. undulata* than to *N. tomentosiformis*, while *N. tabacum* is a hybrid of *N. tomentosiformis* and *N. sylvestris* [69]. Accordingly, in our *GUS* evolutionary tree, two *GUS* copies of *N. tabacum* are present in all clusters, one closely associated with *N. sylvestris* and the other with *N. tomentosiformis*. A similar result (two gene copies of the same species clustering with two different species) is observable with *Brassica napus*, a hybrid of *B. oleracea* and *B. rapa*.

It must also be considered that, after duplication, the gene copies formed begin to diverge. The older the duplication event is, the greater the divergence will be. The indiscriminate use of gene copies can lead to heavy distortions in the phylogenetic tree. For example, using *GUS3.2* for *Nicotiana alata* and *Capsicum annuum* and *GUS3.1* for *N. tabacum* and *Spinacia oleracea* (all named Heparanase3 in database) would lead to locating *N. tabacum* closer to *S. oleracea* than to *N. alata*. Thus, although it requires more work, a preliminary gene evolutionary tree might be useful for the selection of the gene copies which should be used in phylogenetic tree construction.

In addition, we excluded the possibility of alternative splicing events in Arabidopsis, because after PCR sequencing (data not shown) of *AtGUS*s gene transcripts we never found evidence of alternative transcription events.

The analysis of *GUS* evolutionary tree provides also evidence of functional differentiation of baicalinase gene of *Scutellaria baicalensis*. It has been classified as beta-glucuronidase [41,56,70,71], but particular functions have been reported: it is involved in the regulation of hydrogen peroxide production in the cell wall and in the cell apoptosis induced in response to pathogen infection [57,63]. Probably, nucleotide sequence changes accompany functional differentiation, leading to the early divergence of the gene at the roots of the evolutionary tree.

Firstly, concerning the analysis of the evolutionary tree for DFD inference, gene duplication events have been examined in relation to “double-gene phylum” formation and, when it was possible, they have been correlated to known WGD events. *GUS* copies associated with cluster 4 and eudicot 3.2, probably due to ancient duplication events (the first one preceding angiosperm radiation and the second one related to the hexaploidization event in eudicots’ ancestor), are clearly not associated with DFD events, as the originated gene copies have been lost in many species and even in whole families.

At this regard, we underline that the GUS gene expression was detected in all the different tissues of *N. tabacum* analysed, except for GUS4 [72,73]. This suggests that *NtGUS4*, like other short-length *GUS* genes belonging to *GUS3.1* and *GUS4* clusters, undergoes a pseudogenization process. For this reason, we cannot exclude that it is expressed during a specific developmental stage or under different environmental conditions of the organs. Other gene duplication events that occurred in genus or family ancestors are too recent to be strong indicator of DFD events, though the gene copies from which they originate have been conserved in all the derived species. 

No difference in gene expression was observed between AtGUS1 and AtGUS2 expression either in the root [34] or in the other examined part of the plant. However, functional changes cannot be excluded considering that the two GUS1 copies which originated by gene duplication in Brassicaceae ancestor, after Caricaceae diversion, have been conserved in all the examined species.

Arguably, the most important finding of this work is that a DFD event is strongly suggested for *GUS3* and *GUS1*: the two gene copies were derived by the ancient WGD event which occurred in the angiosperm ancestor and both were retained in all the examined extant species. The profound difference in gene expression observed supports the DFD hypothesis suggested by evolutionary gene tree analysis.

### Conclusions and Future Perspectives

We can conclude that our analysis provides useful insights into the broader field of molecular evolution and of different predictive theories concerning environmental changes and plants adaptation. Improved forecasts of plant populations’ adaptive responses under abiotic stress and other environmental pressures require new tools, such as the integrations of the multiple spatial scales of historical and predictive environmental change under modern cohesive approaches such as genomic prediction and machine learning frameworks [74]. Studies on gene evolution for alternative expression patterns in different species are a potential approach to assessing populations’ stress adaptation by looking for favorable alleles. Superior haplotypes/alleles and causal genes for agronomic, climate resilience, and nutrition traits have been traditionally identified via an OMICs approach. As a strategy, using information about different expression patterns could be used in breeding to accelerate the development of superior varieties [75].

Overall, we can conclude that, in the presence of several gene copies, the analysis of the gene evolutionary tree is a useful tool to detect differences in expression or function between gene copies. Besides the theoretical interest in studying differential expression, the possibility exists of using this information in breeding to select favorable alleles. Together with other tools such as genome editing, this will be especially important, in the face of global climatic changes, for the rapid production of climate-resilient crop varieties.

## 4. Materials and Methods

### 4.1. Plant Material: Growth and Fixation

Wild-type *Arabidopsis thaliana* ecotype Columbia (Col-0) seeds used in this study were obtained from The Nottingham *Arabidopsis* Stock Centre (NASC) http://arabidopsis.info (accessed on 1 December 2018).

Seeds of *Arabidopsis thaliana* (L.) Heynh. ecotype Columbia (Col-0) were surface-sterilized by incubation in absolute ethanol for 2 min and 1.75% hypochlorite solution (NaClO) for 12 min. After thorough washing with sterile distilled water (3 × 5 min), the seeds were sown on Petri dishes containing germination medium, 1% sucrose [76] and 0.7% plant cell culture agar. The plated seeds were left at 4 °C for 48 h to ensure uniform germination, and then moved to a growth chamber at 21 °C, under 16 h light (150 μmol m^−2^ s^−1^), 8 h dark and 60% relative humidity.

Root tips, shoot apices and floral buds were collected from 7-, 12- and 20 day-old seedlings, respectively; these were fixed in 4% (*w*/*v*) paraformaldehyde, 15% (*v*/*v*) DMSO and 0.1% Tween 20 in PBS 10% (*w*/*v*), 1 M NaOH, 1 XPBS (10 XPBS: 1.3 M NaCl, 70 mM Na_2_HPO_4_, 30 mM KH_2_PO_4_ pH 7) and chlorophyll was removed, as reported Bruno et al., (2011) [77]. Fixed material was stored in absolute ethanol overnight at −20 °C.

### 4.2. Multiprobe In Situ Hybridization

Fixed whole seedlings were processed for the MISH, as described in Bruno et al., (2015) [34]. *AtGUS1* (AT5G61250) and *AtGUS2* (AT5G07830 riboprobes were generated as described in Bruno et al., (2015) [34]. Concerning AtGUS3 (AT5G34940) the following AtGUS3-FWT7 (5′-CCAAGCTTCTAATACGACTCACTATAGGGAGACTGGACCAAGAGGCAAAAAG-3′) and AtGUS3BW(5′-CTGGACCAAGAGGCAAAAAG-3′) were used in a PCR reaction to amplify gene specific fragment.

Sense and antisense strands were synthesized for each gene by T7 and SP6 RNA polymerase promoters included in the appropriate primer.

Labeled RNA probes were synthesized using in vitro transcription in the presence of Digoxigenin-11-UTP (AtGUS2), Biotin-16-UTP (AtGUS1), and Fluorescein-12-UTP (AtGUS3) and processed as described in Bruno et al., (2015) [34].

### 4.3. Confocal Visualization

Samples were imaged using a Leica inverted TCS SP8 confocal scanning laser microscope with a 40× oil immersion objective. Simultaneous detection of Alexa Fluor dyes (AF) 488, AF 555 and AF 647 was performed by combining the setting indicated in the microscope. The dye conjugates were excited at 488 nm, 555 nm, 647 nm and the fluorescence emissions were assessed at 517 nm, 569 nm and 671, respectively.

### 4.4. Sequence Retrieval from Databases

The sequences of *Arabidopsis thaliana GUS1*, *GUS2* and *GUS3* genes were used as query in nucleotide blast algorithms to the identify conserved GUS gene sequences using BLAST search in National Center for Biotechnology Information (NCBI, https://www.ncbi.nlm.nih.gov/, accessed on 1 May 2022) and Phytozome v12.1 [78] (https://phytozome-next.jgi.doe.gov/, accessed on 1 May 2022) in the default parameter settings.

The research was performed to retrieve all available β-glucuronidase (heparanase) sequences in *Physcomitrella* (Briophyta), in *Selaginella* (Lycophyta) and in angiosperms (Appendix A).

Thus, GUS sequences CDSs were obtained directly from the database. The nucleotide sequences were subsequently aligned with MAFFT (https://mafft.cbrc.jp/alignment/software/, accessed on 1 December 2020) [79] with a gap penalty of 3.0, an offset value of 0.8 and all the default settings. The medium CD sequence length is around 1600 nucleotides (median value). We also discarded: CD sequences shorter than 300 nucleotides; CD sequences shorter than 800 nucleotides and not recognized as heparanases by NCBI or Phytozome v12.1 [78]; and CD sequences determined in the same species and in the same locus.

CD sequence length, reference number, chromosome and locus are reported in Appendix A. In order to make the analysis of the data easier, the name of the genes has been changed according to their position in the evolutionary tree.

### 4.5. Evolutionary Tree Construction

Due to the high number of CDSs, preliminary *GUS* evolutionary trees were constructed at family level (data not shown). To simplify the overall analysis, the gene copies of the same species tightly clustered within the family were not used for the construction of the angiosperm evolutionary tree. In Appendix A, the CDSs used for evolutionary tree construction are reported in bold.

CDSs alignment was performed using the GUIDANCE2 algorithm [80] with default parameters [81]. Tree reconstruction was performed using the maximum likelihood algorithm in the software package MEGA 10 [82]. The alignment was resampled by the bootstrap method 1000 times.

## Figures and Tables

**Figure 1 plants-12-02051-f001:**
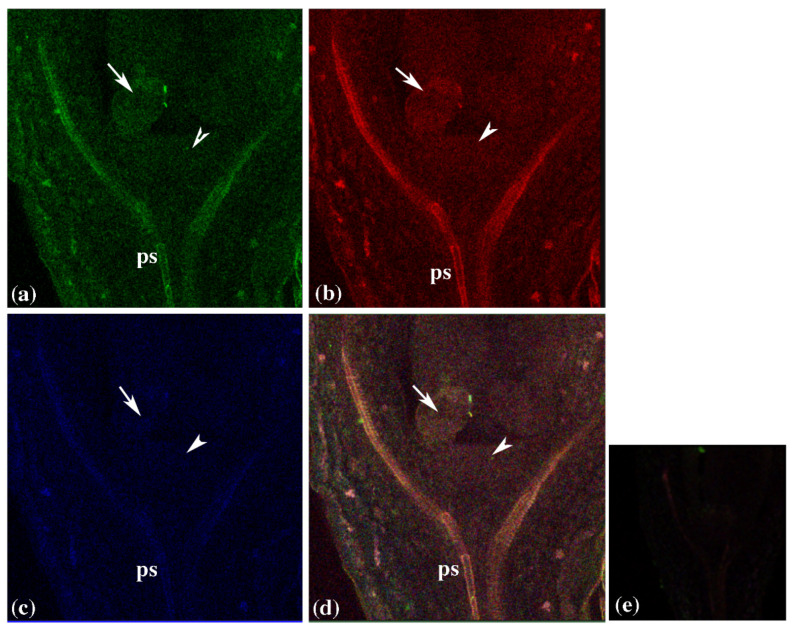
Triple-labelled whole-mount fluorescent in situ hybridization of *AtGUS genes* in *Arabidopsis thaliana* shoot apex. (**a**) *AtGUS1* BIO riboprobe, mouse anti-BIO and AF488 donkey anti-mouse (green); (**b**) *AtGUS2* DIG riboprobe, sheep anti-DIG and AF555 donkey anti-sheep (red); (**c**) *AtGUS3* FITC riboprobe, rabbit anti-FITC and AF647 chicken antirabbit (blue); (**d**) merge; (**e**) control sense. Key: DIG, digoxigenin; BIO, biotin; FITC, fluorescein; AF, Alexa Fluor dyes. Set. Scale bars, (**a**,**e**), 75 μm. Ps = provascular strand. White arrowheads indicate the signal detected in the apical dome leaf primordia and the white arrows indicate the signal detected in the emerging leaflets.

**Figure 2 plants-12-02051-f002:**
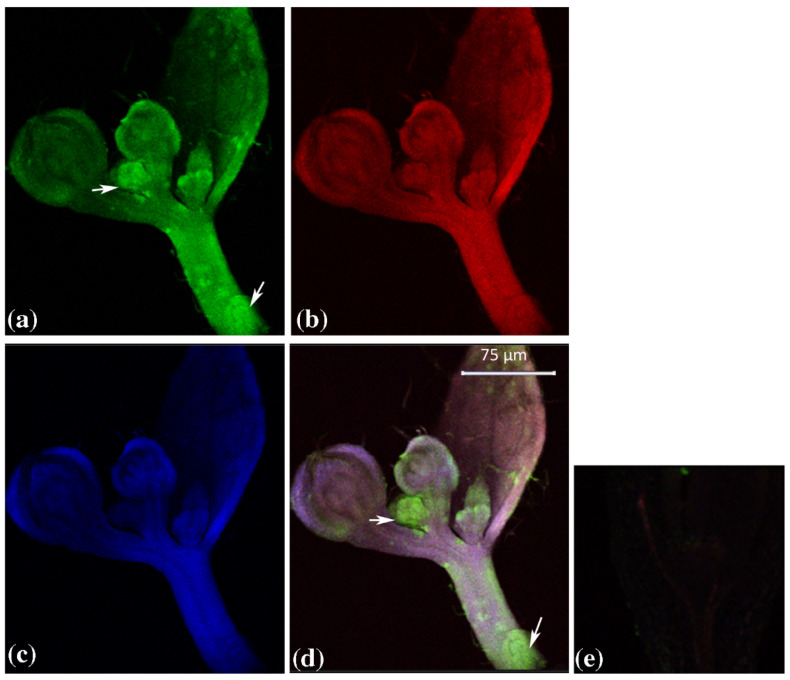
Triple-label whole-mount fluorescent in situ hybridization of AtGUS genes in *Arabidopsis thaliana* floral bud. Green: (**a**) AtGUS1 BIO riboprobe, mouse anti-BIO and AF488 donkey anti-mouse; Red: (**b**) AtGUS2 DIG riboprobe, sheep anti-DIG and AF555 donkey anti-sheep; Blue: (**c**) AtGUS3 FITC riboprobe, rabbit anti-FITC and AF647 chicken anti-rabbit; Merge (**d**). (**e**) control sense. Key: DIG, digoxigenin; BIO, biotin; FITC, fluorescein; AF, Alexa Fluor dyes. Set. Scale bars, (**a**,**e**), 75 μm. White arrows indicate the signal detected in the floral organ blastozone.

**Figure 3 plants-12-02051-f003:**
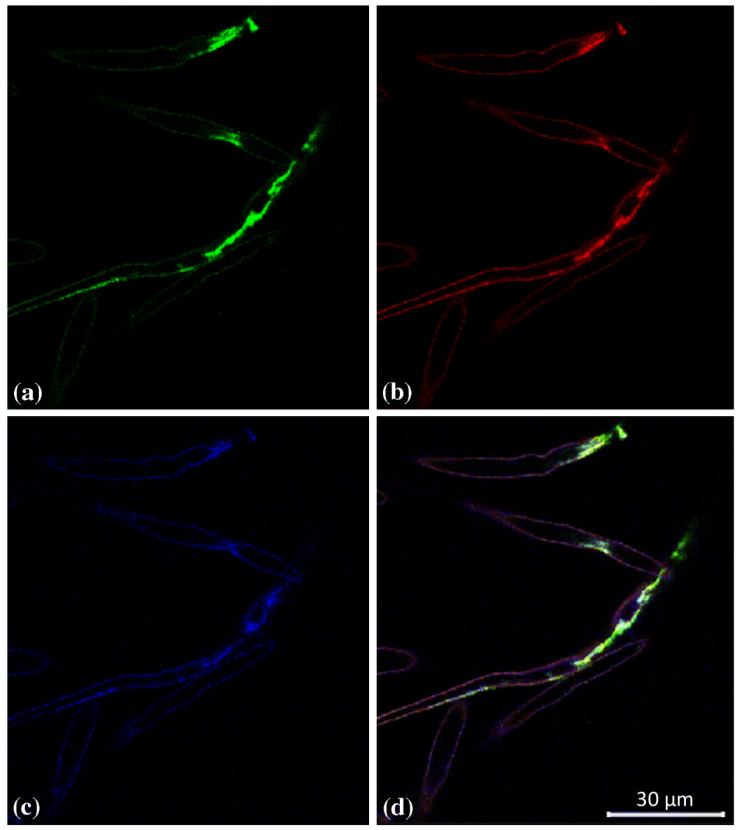
Triple-label whole-mount fluorescent in situ hybridization of *AtGUS genes* in *Arabidopsis thaliana* root hair. Green: (**a**) *AtGUS1* BIO riboprobe, mouse anti-BIO and AF488 donkey anti-mouse; Red: (**b**) *AtGUS2* DIG riboprobe, sheep anti-DIG and AF555 donkey anti-sheep; Blue: (**c**) *AtGUS3* FITC riboprobe, rabbit anti-FITC and AF647 chicken antirabbit; Merge (**d**). Control sense key: DIG, digoxigenin; BIO, biotin; FITC, fluorescein; AF, Alexa Fluor dyes, (Molecular Probes). Set. Scale bars, (**a**,**d**), 75 μm.

**Figure 4 plants-12-02051-f004:**
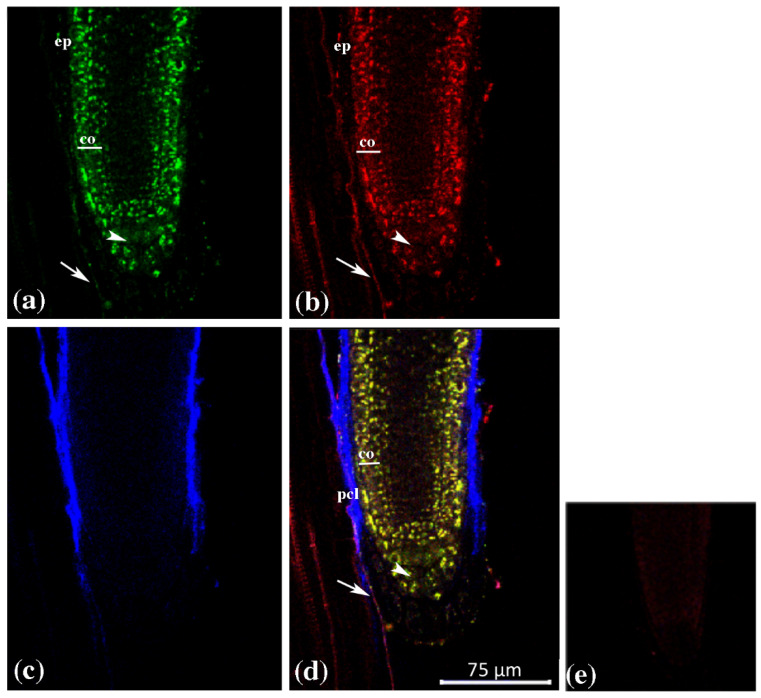
Triple-label whole-mount fluorescent in situ hybridization of *AtGUS genes* in *Arabidopsis thaliana* root apex. Green: (**a**) *AtGUS1* BIO riboprobe, mouse anti-BIO and AF488 donkey anti-mouse; Red: (**b**) *AtGUS2* DIG riboprobe, sheep anti-DIG and AF555 donkey anti-sheep; Blue: (**c**) *AtGUS3* FITC riboprobe, rabbit anti-FITC and AF647 chicken anti-rabbit; Merge (**d**). (**e**) control sense Key: DIG, digoxigenin; BIO, biotin; FITC, fluorescein; AF, Alexa Fluor dyes. Set. Scale bars, (**a**,**e**), 75 μm. co = cortex; ep = epidermis; plc = protodermal cell line. White arrowheads indicate the signal detected in the root cap meristem and the white arrows indicate the signal detected in the cortical cell layers of proximal meristem.

**Figure 5 plants-12-02051-f005:**
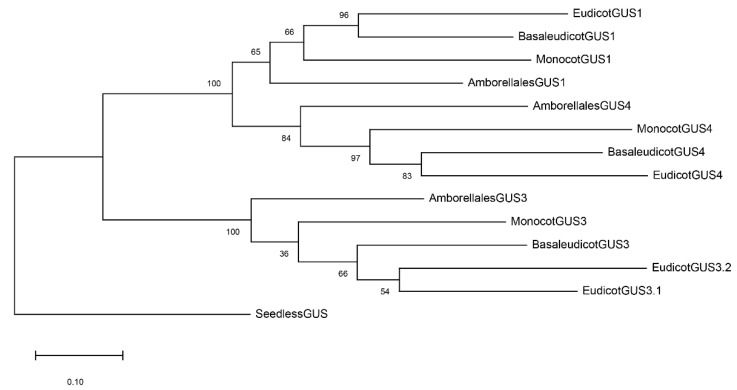
Schematic representation of the evolutionary tree of GUS genes in plants. The following 14 full-length cDNA sequences have been used to build the tree by maximum likelihood (bootstrap = 500): *Physcomitrella patens* (PhypatGUS1.a) for seedless GUS; *Amborella trichopoda* (AmbtriGUS1, AmbtriGUS3 and AmbtriGUS4) for Amborelalles; *Ananas comosus* (AnacomGUS1.1, AnacomGUS3.a and AnacomGUS4) for monocots; *Aquilegia caerulea* (AqucaeGUS1, AqucaeGUS3, AqucaeGUS4) for basal eudicots; *Nicotiana sylvestris* (NicsylGUS1.1, NicsylGUS3.1, NicsylGUS3.2, NicsylGUS4) for eudicots.

## Data Availability

Data is contained within the article and in the Appendix A.

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
