# Peer review of "A Study of GUS Expression in *Arabidopsis* as a Tool for the Evaluation of Gene Evolution, Function and the Role of Expression Derived from Gene Duplication"

_plants, 2023, doi:10.3390/plants12102051_

Round 1

Reviewer 1 Report

In this manuscript, Bruno at al use multiprobe in situ hybridization (MISH) to look for potential differences in the expression pattern of 3 Arabidopsis thaliana GUS genes, GUS1, GUS2, and GUS3. They identified differences in GUS3 root expression and explain this by a possible 'duplication plus functional differentiation' (DFD) event on the two copies derived from a whole genome duplication (WGD) occurred in the ancestor of the Angiosperms.

The text is quite convoluted and confusing because of a strange sentence construction which makes the ideas somewhat obscure. The results section is also exceedingly repetitive, I felt like I was reading the same paragraph multiple times and I think that also contributed to diluting the main point the authors were trying to convey.

In lines 133-135 the authors state their main goal with this work, which was to infer function from the evolutionary tree of gene families that have retained multiple copies, and specifically of GUS. This goal, however, is not tested by the proposed methodology.

In the discussion, the authors use their data to infer on Nicotiana tabacum GUS4, and other genes without providing any data to support it. Perhaps it is a prediction of their model, but again, the text is obscure.

Reviewer 2 Report

This manuscript by Bruno et al. studied GUS expression in Arabidopsis in order to test whether gene function and expression change after duplication. The work is hypothesis-driven, well motivated and planned, and targets a key expectation on gene evolution. However, I am afraid the transcriptomic sampling applied in this study may have low power to detect rare alternative splicing events in Arabidopsis, which may also occur after gene duplication. Authors need to acknowledge this caveat, and discuss whether a potential lack of power may tend ignoring rare alternative splicing events. Besides, is there certainty that inferred expression profiles are not contaminated with paralogous variation? I encourage authors to comment on these analyses and discuss this in the manuscript. Some other aspects that require attention are listed here:      

- Include explicit research questions and expected results in the last paragraph of the introduction (L136).

- How were candidate WUS genes optimized in the light of recombination rate variation and linkage disequilibrium across the genomes? If two candidates were close by, they may not be independent enough for testing. Please comment in the light of the linkage disequilibrium patterns (refer to PLoS One 2018 13(3):e0189597? Besides, are genes at low recombining regions less prompt to exhibit alternate features (refer to Front Plant Sci 2018 9:1816)? In order to address these relevant points, please add a figure marking the genetic map of the studied genes and their paralogous.

- Take into account that it is never beyond the scope of any research to explicitly acknowledge the study caveats, especially when dealing with sometimes confounding alternative expression profiles. Therefore, short perspectives and conclusion sections (in L393) would be insightful for readers to fill the identified caveats as part of future research when addressing functional evolutionary diversity in terms of gene evolution.

- Please also embrace in a novel perspectives section at L394 how may the identified alternative expression patterns source gene evolution studies in other species? Specifically, although the report provides basic evidence into the expression diversity of WUS genes, a major question that authors should prospect in their discussion is how to unlock and effectively utilize these novel gene functionalities. Speed breeding, gene editing, recurrent backcrossing and inter-specific schemes (condensed, contrasted and discussed in Trends Genet 2021 37:1124-1136, and Front Genet 2020 11:564515, please include) may offer some insights.

Round 2

Reviewer 2 Report

Authors have addressed the review suggestions accordingly, and properly argued against some of these in the rebuttal. The work is valuable because it explicitly tests a hypothesis on gene functionality as part of evolutionary duplication events. Therefore, from my side it is feasible to proceed with the divulgation of this report.
